# Denoised Smoothing with Sample Rejection for Robustifying Pretrained Classifiers

**Fatemeh Sheikholeslami**[1] *    **Wan-Yi Lin**[2]    **Jan Hendrik Metzen**[2]
**Huan Zhang**[3]    **Zico Kolter**[2,3]

[1] Amazon    [2]Bosch Center for AI    [3] Carnegie Mellon University
{shfateme}@amazon.com
{wan-yi.lin}@us.bosch.com
{janhendrik.metzen}@de.bosch.com
{huan}@huan-zhang.com
{zkolter}@cmu.cs.edu

## Abstract

Denoised smoothing is the state-of-the-art approach to defending pretrained classifiers against $\ell_p$ adversarial attacks, where a denoiser is prepended to the pretrained classifier, and the joint system is adversarially verified via randomized smoothing. Despite its state-of-the-art certified robustness against $\ell_2$-norm adversarial inputs, the pretrained base classifier is often quite uncertain when making its predictions on the denoised examples, which leads to lower natural accuracy. In this work, we show that by augmenting the joint system with a "rejector" and exploiting adaptive sample rejection, (i.e., intentionally abstain from providing a prediction), we can achieve substantially improved accuracy (especially natural accuracy) over denoised smoothing alone. That is, we show how the joint classifier-rejector can be viewed as a classification-with-rejection per sample, while the smoothed joint system can be turned into a robust *smoothed classifier without rejection*, against $\ell_2$-norm perturbations while retaining certifiability. Tests on CIFAR10 dataset show considerable improvements in *natural* accuracy without degrading adversarial performance, with affordably-trainable rejectors, specially for medium and large values of noise parameter $\sigma$.

## 1   Introduction

Despite their success in image classification, deep learning models are known to be vulnerable against $\ell_p$-*norm adversarial attack*, where small imperceptible perturbations on the input image can considerably change model predictions Biggio et al. [2013], Szegedy et al. [2013], Goodfellow et al. [2014], Carlini and Wagner [2017], Uesato et al. [2018]. Many empirical defense mechanisms and training procedures have been proposed against adversarial attacks while often times stronger attacks have followed to break them [Athalye et al., 2018, Tramèr et al., 2017]. These advances have lead to certifiable defenses Levine and Feizi [2020], Wong and Kolter [2018], Salman et al. [2020], Gowal et al. [2018] which provide provable lower bounds of robust classification accuracy; the most relevant to our work is randomized-smoothing Lecuyer et al. [2019], Cohen et al. [2019], Li et al. [2018].

*Denoised smoothing* Salman et al. [2020] approaches this task by prepending a denoiser to *pretrained* classifiers, and utilizes it jointly with randomized smoothing, i.e., taking a majority vote of the predictions from multiple copies of the input image with noise, and provides state-of-the-art certifiable defense for pretrained classifiers for $\ell_2$-norm perturbations. Unfortunately, despite using pretrained

---

*Author was with Bosch Center for AI when this work was done.

2022 Trustworthy and Socially Responsible Machine Learning (TSRML 2022) co-located with NeurIPS 2022.

classifiers with high accuracy, the resulting smoothed classifier still exhibits a substantial drop in clean accuracy for affordably-trainable denoisers. On the other hand, utilization of a rejection class for improving robustness is an open area of research [Geifman and El-Yaniv, 2019, Yin et al., 2019, Stutz et al., 2020, Liu et al., 2019, Tramer, 2022], with limited work available with certifiable guarantees against $\ell_\infty$-norm perturbations Sheikholeslami et al. [2020]. In fact, a certifiable utilization of detector/rejectors has shown to be necessary as carefully-designed adaptive attacks have frequently broken existing non-verifiable detectors Athalye et al. [2018], Tramer et al. [2020].

**Our contribution**.

To the best of our knowledge, this is the first work to robustify pretrained classifiers with sample rejection while providing certifiable accuracy. Key to our approach is to use a reject class, realized through *cheaply-trainable per-class rejectors*, which are trained to reject noisy samples whose prediction is inconsistent with the prediction of the clean sample. Inevitably this can also lead to a small number of correctly classified samples to also get rejected, however, the overall certification radius with a pretrained denoiser is improved since: (a) the reject class is used to provide a lower (and tighter) upper bound on the wrong class probabilities- see Fig. 1- and subsequently (b) the lowered probability of the runner-up class leads to higher certification radius due to its non-linear dependence via the inverse Gaussian CDF function. See Fig. 2 for a schematic of the proposed joint system.

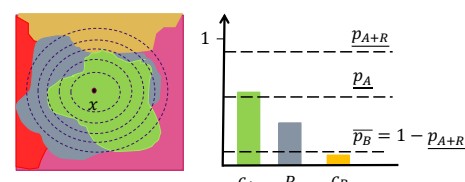

Figure 1: Verification of the proposed smoothed classifier with 'sample rejection' at input x. **Left**: Different colors denote the decision regions of the base classifier $f_R(.)$ for different classes, with color grey denoting the 'reject' class. The dotted lines are the level sets of the Gaussian noise distribution. **Right**: lower bound on top-class probability $\underline{p_A}$ and that of "top-or-reject-class" $p_{A+R}$, used for upperbounding the probability of every other class $\overline{p_B}$.

## 2 Background and Related Work

### 2.1 Randomized smoothing

Consider a classification problem from $\mathbb{R}^d$ to classes $\mathcal{Y} := \{1, 2, \cdots, K\}$. One can construct a "smoothed" classifier $g$ from an arbitrary base classifier $f$ by defining [Cohen et al., 2019] $g(x) := \arg\max_{c \in \mathcal{Y}} \pi_c$ where $\pi_c := \mathbb{P}(f(x + \epsilon) = c)$ and $\epsilon \sim \mathcal{N}(0, \sigma^2 I)$. That is, the smoothed classifier $g$ returns the class that the base classifier $f$ is most likely to return if the input $x$ is perturbed under Gaussian Noise $\epsilon \sim \mathcal{N}(0, \sigma^2 I)$. Let us also denote bounds on class probabilities by $\underline{\pi_c} \leq \pi_c \leq \overline{\pi_c}$.

**Certification:** The main advantage of the well-known randomized-smoothing method is its inherent capability in providing certifiable robustness against bounded $\ell_2$-norm worst-case perturbations. Formally, for any deterministic or random function $f : \mathbb{R}^d \to \mathcal{Y}$, suppose $c_A, c_B \in \mathcal{Y}$ are the top and runner-up class respectively, and $\underline{\pi_A}, \overline{\pi_B} \in [0, 1]$ satisfy:

$$\mathbb{P}(f(x + \epsilon) = c_A) \geq \underline{\pi_A} \geq \overline{\pi_B} \geq \max_{c \neq c_A} \mathbb{P}(f(x + \epsilon) = c).$$

Then Cohen et al. [2019] proved a tight verification bound as follows: $g(x + \delta) = c_A$ for all $\|\delta\|_2 < \rho$, where $\rho = \dfrac{\sigma}{2} \left( \Phi^{-1}\left(\underline{\pi_A}\right) - \Phi^{-1}\left(\overline{\pi_B}\right) \right)$ and $\Phi^{-1}(.)$ is the inverse of the standard Gaussian CDF. In practice, Monte Carlo (MC) sampling is used to estimate the class $c_A$ and a lower bound on its class probability $\underline{\pi_{A(MC)}}$. Using $\pi_B \leq 1 - \underline{\pi_{A(MC)}}$ yields certification radius $\rho \geq \sigma \Phi^{-1}(\underline{\pi_{A(MC)}})$.

### 2.2 Denoised smoothing for defending pretrained classifiers

Despite its simplicity, randomized smoothing is not, in general, directly effective on pretrained classifiers. Specifically, performance of an off-the-shelf classifier can considerably deteriorate when the input is subject to Gaussian noise (leading to small $\pi_A$, and subsequently small certification radius $R$), as standard classifiers, in general, are not trained to be robust against Gaussian perturbations.

In order to construct robust classifiers without altering the underlying weights of a given network $f(.)$, Salman et al. [2020] proposed to use an image denoiser as a pre-processing step before

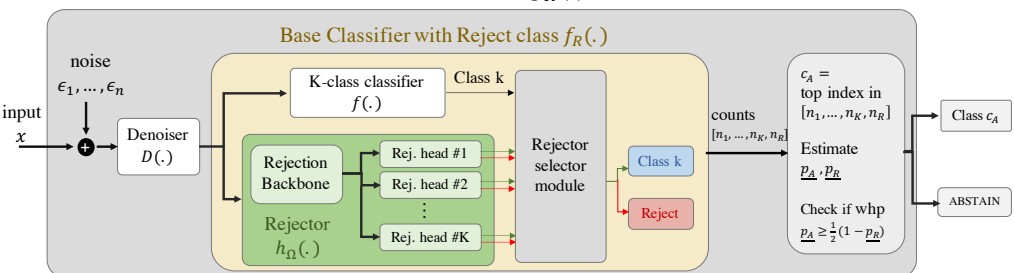

Figure 2: Schematic of the overall robust system consisting of the pretrained classifier $f$ and denoiser $D$, and trainable rejectors $\{h_k\}_{k=1}^K$.

passing inputs through $f(.)$, where the denoiser aims at removing the Gaussian noise added to the input. Concretely, this is done by augmenting the classifier $f$ with a custom-trained denoiser $D_\theta(.) : \mathbb{R}^d \to \mathbb{R}^d$, rendering the entire system as the composite function $f \circ D_\theta : \mathbb{R}^d \to \mathcal{Y}$. Such denoisers can be trained using various objectives subject to a varying level of complexity. Minimizing the mean-square-error (MSE) loss of the reconstructed image as the simplest option and the "stability" loss as a more expensive (to train) option are proposed, formulated respectively as $L_{\text{MSE}}(\theta) = \mathbb{E}_{x_i, \epsilon} \left[ \|D_\theta(x_i + \epsilon) - x_i\|_2^2 \right]$ and $L_{\text{stability}}(\theta) = \mathbb{E}_{x_i, \epsilon} \left[ \ell_{\text{CE}} \left( F \left( D_\theta \left( x_i + \epsilon \right) \right), f \left( x_i \right) \right) \right]$ where $\theta$ denotes trainable parameters of the denoiser, and the expectation is taken over data and noise. Utilizing $L_{\text{stability}}$ has proven to be successful, while best performance is achieved by imposing up-to an order of magnitude increase in training time and complexity compared to using the MSE loss.

# 3 Denoised smoothing with sample rejection

In this work, we aim to improve certification accuracy of pretrained classifiers by incorporating an explicit 'reject' class into the base classifier, while preserving the certifiability against worst-case perturbations with bounded $\ell_2$-norm. To this end, let $h : \mathbb{R}^d \to \{0, 1\}$ denote a general function with binary outputs, which effectively 'flags' the input $x$ if $h(x) = 1$, thus assigning it to the reject class; while $h(x) = 0$ indicates allowing the input to pass and thus not rejecting it. In this work, we introduce a novel algorithm to effectively *train* and *operate* such a 'rejector' in conjunction with pretrained denoised smoothing in order to improve the robust performance of a pretrained classifier.

**Base classifier with a rejection class:** Let us denote $K$ distinct binary classifiers by $\{h_k\}_{k=1}^K$, which we also refer to as rejectors in this work. Together with the base classifier $f$, our proposed *base-classifier-with-rejection* $f_R(x) : \mathbb{R}^d \to \mathcal{Y}_+$, when queried at $x$, returns one of the classes in $\mathcal{Y}_+ := \mathcal{Y} \cup \{R\}$, where class $R$ denotes the reject class. Let us now formally define $f_R(.)$.

**Definition**. Consider $f : \mathbb{R}^d \to \mathcal{Y}$ to be any given deterministic or random classifier, together with a collection of $K$ deterministic or random *binary* classifiers, denoted by $\{h_k\}_{k=1}^K$ where $h_k : \mathbb{R}^d \to \{0, 1\}$. The *base-classifier-with-rejection* $f_R(x) : \mathbb{R}^d \to \mathcal{Y}_+$ is defined as

$$f_R(x) := (1 - h_{f(x)}(x))f(x) + h_{f(x)}(x)R = \begin{cases} k & \text{if } f(x) = k \text{ and } h_k(x) = 0 \\ R & \text{if } f(x) = k \text{ and } h_k(x) = 1 \end{cases} \tag{1}$$

**Smoothed classifier (without a rejection class)** For a given base classifier $f_R(.)$, define

$$g_R(x) := \arg\max_{c \in \mathcal{Y}} \mathbb{P}\left[ f_R(x + \epsilon) = c \right], \tag{2}$$

where noise $\epsilon$ is sampled from Gaussian distribution with variance $\sigma$, i.e., $\epsilon \sim \mathcal{N}(0, \sigma^2 I)$. That is, among the original classes in $\mathcal{Y}$, the smoothed classifier returns the class that is mostly likely to be returned by the base-classifier-with-rejection $f_R(.)$, *excluding* the rejection class $R$.

## 3.1 Certified Robustness

We prove that the smoothed classifier $g_R(.)$ defined in 2 is robust within the $\ell_2$ radius in Theorem 3.1.

**Theorem 3.1.** *Let $f_R : \mathbb{R}^d \to \mathcal{Y}_+$ be any deterministic or random function, $\epsilon \sim \mathcal{N}(0, \sigma^2 I)$, and let $g_R(.)$ be defined as in Eq. 2. Respectively, denote top and runner-up class probabilities (exc. class R) as $p_A := \mathbb{P}(f_R(x + \epsilon) = c_A)$, $p_B := \max_{c \notin \{c_A, R\}} \mathbb{P}(f_R(x + \epsilon) = c)$, and define $p_{A+R} := \mathbb{P}(f_R(x + \epsilon) \in \{c_A, R\})$. Suppose for $c_A \in \mathcal{Y}$, lower bounds $\underline{p_A}, \underline{p_{A+R}} \in [0, 1]$ satisfy $p_A \geq \underline{p_A} \geq p_B$ and $p_{A+R} \geq \underline{p_{A+R}}$. Then $g_R(x + \delta) = c_A$ for all $\|\delta\|_2 \leq \rho$, where*

$$\rho = \frac{\sigma}{2}\left(\Phi^{-1}\left(\underline{p_A}\right) + \Phi^{-1}\left(\underline{p_{A+R}}\right)\right)$$

Details of proof is delegated to Appendix 6.1 for brevity. It is important to note that the claim of this Theorem is tightly connected to the results in [Cohen et al., 2019]; while here by utilizing the rejection class $R$, the upperbound $p_B \leq 1 - \underline{p_{A+R}}$ is employed. Furthermore, the intuition behind why this can potentially lead to better certificatino radius is that proper training and expressiveness of the rejectors can lead to $p_A + p_R > \pi_A$ and, owing to the non-linearity of the inverse CDF especially at large values, can lead to an improved radius if

$$\frac{\Phi^{-1}(\underline{p_{A,(MC)}}) + \Phi^{-1}(\underline{p_{A+R,(MC)}})}{2} \geq \Phi^{-1}(\underline{\pi_{A,(MC)}}).$$

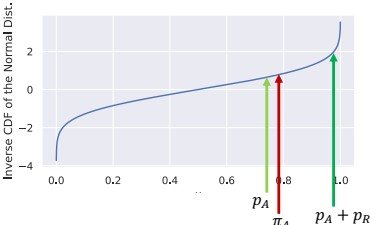

Figure 3: Inverse CDF of a standard Gaussian distribution.

## 3.2 Sample rejection with denoised smoothing

The performance gap in empirical results between what an affordably trained denoiser and that of the robustly trained baseline implies the limited capability of the joint denoiser-classifier system in correctly classifying denoised inputs especially those subject to large noise. To alleviate the performance loss in such cases, we propose to use the rejection capability in order to improve network accuracy. Concretely, we aim at utilizing rejectors $\{h_k\}_{k=1}^K$ in blocking inputs likely to be mis-classified by the following overall classification system:

$$g_R(x) := \arg\max_{c \in \mathcal{Y}} \mathbb{P}\Big[f_R(D(x + \epsilon)) = c\Big] \quad \text{where} \quad \epsilon \in \mathcal{N}(0, \sigma^2 I). \tag{3}$$

**Practice**. During inference, the image $x$ will go through the following steps: (a) it is first perturbed by noise $\epsilon$, (b) passes through the image preprocessing (denoising) step via $D(x + \epsilon)$, and (c) the resulting denoised image goes through $f_R(D(x + \epsilon))$ as in Eq. (1); (d) finally, the classification output of the overall system $(D, f, \{h_k\}_{k=1}^K)$ is claimed as the most likely class over the noise distribution (or it empirical realization via $N$ iid samples). The schematic in Fig. 2 depicts a visual placement of the components $\mathcal{S} = \{D, f, \{h_k\}_{k=1}^K\}$. Algorithm 1, 2 and 3 provide the pseudocode for the prediction and certification (via MC-based class probability

---

**Algorithm 1** Prediction

PREDICT $(\mathcal{S} = \{D, f, \{h_k\}_{k=1}^K\}, \sigma, x, N, \alpha)$
    count $\leftarrow$ SAMPLEUNDERNOISE $(\mathcal{S}, \sigma, x, N)$
    $\widehat{c_A}, \widehat{c_B} \leftarrow$ top two indices in count
    $n_A, n_B \leftarrow$ count $[\widehat{c_A}]$, count$[\widehat{c_B}]$
    **if** BINOMPVALUE$(n_A, n_A + n_B, 0.5) \leq \alpha$
    **then**
        return $\widehat{c_A}$
    **else**
        return ABSTAIN
    **end if**

---

lowerbounds) of the overall system, while training the rejectors in a scalable and stable manner is detailed in Appendix 6.2.

Condition $p_A > p_B$ is substituted by the more restrictive condition $p_A \geq \underline{p_A} \geq 1 - \underline{p_A} - \underline{p_R} \geq p_B$ which leads to $\underline{p_A} > \frac{1}{2}(1 - \underline{p_R})$, which if not met, the algorithm abstains from certification. This is to be contrasted with $\pi_A > 1/2$ in randomized smoothing [Cohen et al., 2019]. Function LOWERCONFBOUND$(s, n_0, 1 - \alpha)$ returns a one-sided $(1 - \alpha)$ lower confidence interval for the Binomial parameter $q$ given a sample $s \sim \text{Binomial}(n_0, q)$, which similar to Cohen et al. [2019], has been evaluated via Clopper–Pearson interval in the *statsmodel* package in Python directly. Subscript MC has been dropped in Algorithm 2 for brevity of notation.

**Algorithm 2** Certification with MC-sampling

*# certify the robustness of $g_R$ around $x$*
CERTIFY $(\mathcal{S} = \{D, f, \{h_k\}_{k=1}^K\}, \sigma, x, N, \alpha)$
    count $\leftarrow$ SAMPLEUNDERNOISE $(\mathcal{S}, \sigma, x, N)$
    $\widehat{c_A} \leftarrow$ top index in count
    $n_A, n_R \leftarrow$ count $[\widehat{c_A}]$, count $[R]$
    $\underline{p_A} \leftarrow$ LOWERCONFBOUND$(n_A, N, 1 - \alpha)$
    $\underline{p_R} \leftarrow$ LOWERCONFBOUND$(n_R, N, 1 - \alpha)$
    $\underline{p_{A+R}} \leftarrow$ LOWERCONFBOUND$(n_A + n_R, N, 1 - \alpha)$
    **if** $\underline{p_A} > \dfrac{1}{2}(1 - \underline{p_R})$ **then**
        return $\widehat{c_A}$ , $\hat{\rho} = \dfrac{\sigma}{2}(\Phi^{-1}(\underline{p_A}) + \Phi^{-1}(\underline{p_{A+R}}))$
    **else**
        return ABSTAIN
    **end if**

**Algorithm 3** Sampling under noise for the overall system $\mathcal{S} = \{D, f, \{h_k\}_{k=1}^K\}$

    **function** SAMPLEUNDERNOISE $(\mathcal{S}, x, n, \sigma)$
    Initialize count $= [0, \cdots, 0]_{K+1 \times 1}$
    **for** $\nu = 1, \cdots, n$ **do**
        sample noise $\epsilon_\nu \in \mathcal{N}(0, \sigma^2 \mathbf{I})$
        $k \leftarrow f(D(x + \epsilon_\nu))$
        **if** $h_k(D(x + \epsilon_\nu)) = 0$ **then**
            ++ counts $[k]$
        **else**
            ++ counts $[R]$
        **end if**
    **end for**
    return count

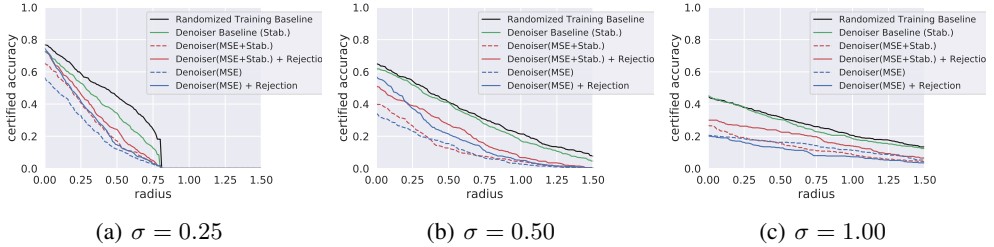

(a) $\sigma = 0.25$          (b) $\sigma = 0.50$          (c) $\sigma = 1.00$

Figure 4: CIFAR10 certification results with DnCNN-based denoising

(a) $\sigma = 0.25$          (b) $\sigma = 0.50$          (c) $\sigma = 1.00$

Figure 5: CIFAR10 certification results with MemNet-based denoising

## 4 Experiments

In this section, we examine the performance of robustifying pretrained classifier using rejectors with denoised smoothing on CIFAR10 dataset with ResNet110 as the base classifier, and comparing our method with two baselines: randomized smoothing (serving as the upper-bound baseline) and denoised smoothing (with two choices of denoiser architectures, namely DnCNN [Zhang et al., 2017] and MemNet [Tai et al., 2017]), both of which have network weights publicly available. To test the proposed Sample rejection with denoised smoothing, the classifier and denoiser architectures and weights are considered given and fixed. Rejector network is implemented with a ResNet-34-like backbone architecture with a 10-dimensional output, to realize the $K = 10$ rejector heads, trained with 20 epochs and SGD optimizer; see App. 6.3 for detailed description of architectures and training.

Performance is reported in terms of certified accuracy in 4 and 5, scatter plots in 6, and training time in 1 demonstrating the effective exploitation of the rejectors. Although certification accuracy of our method is lower than expensive denoised smoothing baseline with stability objectives, our training time is less than half of training a denoiser with stability loss – training a denoisier with MSE or MSE and stability objective in addition to train the rejector is in total 2.78 to 6.8 hours while training a denoiser with stability objective is 9.8 to 20.8 hours. Also, 6 plots the 2D-histograms corresponding

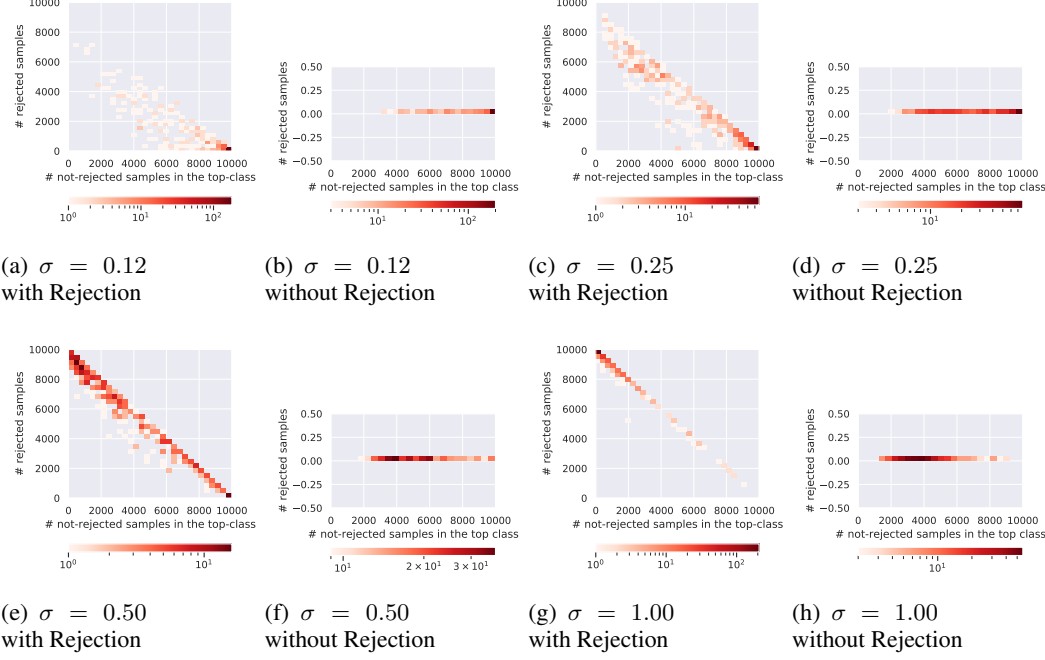

(a) $\sigma = 0.12$
with Rejection

(b) $\sigma = 0.12$
without Rejection

(c) $\sigma = 0.25$
with Rejection

(d) $\sigma = 0.25$
without Rejection

(e) $\sigma = 0.50$
with Rejection

(f) $\sigma = 0.50$
without Rejection

(g) $\sigma = 1.00$
with Rejection

(h) $\sigma = 1.00$
without Rejection

Figure 6: 2D-histogram plots of distribution of the joint classification-rejection outcome in terms of number-of-rejected vs. not-rejected top-class samples. Utilizing the rejection class helps the summation of the two terms to approach the total number of N=10,000 draws (the $x_1 + x_2 = N$ diagonal line), which can help with improving the certification radius as discussed in Remark.

to the overall denoiser-classifier-rejector system in terms of number-of-rejected vs. top-class not-rejected samples for DnCNN denoiser trained with the MSE-loss and fine-tuned with the stability loss, exhibiting the utilization of the rejection class for large values of $\sigma$ while demonstrating that the summation of the two values approaches the total number of $N = 10,000$ draws.

Table 1: Training time of different denoisers and rejectors. '+' signs mean training time is in addition to the previous rows' times/epochs since they refer to fine-tuning and/or the rejectors' training times.

|  | Objective | Epochs | Sec per Epoch | Total Time (hr) |
|---|---|---|---|---|
| DnCNN | MSE | 90 | 31 | 0.78 |
| DnCNN | +Stab | +20 | 57 | +0.32 |
| DnCNN + Rejector | +Reject. Obj. | + 20 | 352 | +1.94 |
| DnCNN | Stab. Obj. | 600 | 59 | 9.80 |
| MemNet | MSE | 90 | 85 | 2.13 |
| MemNet | +Stab | +20 | 118 | +0.66 |
| MemNet + Rejector | +Reject. Obj | +20 | 546 | +3.03 |
| MemNet | Stab. Obj. | 600 | 125 | 20.83 |

## 5 Conclusions

In this work, we have proposed a novel algorithm for modeling and utilizing an 'explicit reject' class for robustifying pre-trained classifiers with denoised smoothing with certifiable guarantees. The reject class, effectively realized through *cheaply-trainable per-class rejectors*, is successfully exploited through MC sampling to reject noisy samples whose prediction is inconsistent with the prediction of the clean sample. Thus, by reducing the number of not-rejected misclassified samples, natural accuracy as well as the certification radius is shown to have substantially improved for inexpensive denoisers. To the best of our knowledge, this is the first work to incorporate certifiable application of rejection for robustifying pre-trained classifiers with denoised smoothing.

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

# 6 Appendix

## 6.1 Proof of Theorem 3.1

**Theorem** [Restated] *Let $f_R : \mathbb{R}^d \to \mathcal{Y}_+$ be any deterministic or random function, $\epsilon \sim \mathcal{N}(0, \sigma^2 I)$, and let $g_R(.)$ be defined as in Eq. 2. Suppose $c_A \in \mathcal{Y}$ and $\underline{p_A}, \underline{p_{A+R}} \in [0, 1]$ satisfy*

$$\mathbb{P}(f_R(x + \epsilon) = c_A) \geq \underline{p_A} \geq \max_{c \neq c_A, R} \mathbb{P}(f_R(x + \epsilon) = c)$$

$$\mathbb{P}(f_R(x + \epsilon) \in \{c_A, R\}) \geq \underline{p_{A+R}}.$$

*Then $g_R(x + \delta) = c_A$ for all $\|\delta\| \leq R$, where*

$$R = \frac{\sigma}{2} \left( \Phi^{-1} \left( \underline{p_A} \right) + \Phi^{-1} \left( \underline{p_{A+R}} \right) \right).$$

*Proof.* We need to prove that for $\|\delta\|_2 \leq R$, the classification outcome of $g_R(.)$ will not flip from $c_A$ to any other class. According to the definition of $g_R(.)$ in Eq. (2), class $R$ is never selected as its outcome, thus it suffices to show

$$\mathbb{P}(f_R(x + \delta + \epsilon) = c_A) \geq \mathbb{P}(f_R(x + \delta + \epsilon) = c) \quad \forall c \notin \{c_A, R\} \tag{4}$$

where the probability is computed over the randomness of noise $\epsilon$. As a direct result from the certification guarantee in Cohen et al. [2019], this would hold for

$$R = \frac{\sigma}{2}(\Phi^{-1}(p_A) - \Phi^{-1}(p_B)) \tag{5}$$

where $\overline{p_B} \geq p_B = \max_{c \notin \{c_A, R\}} \mathbb{P}(f_R(x + \delta) = c)$ and $c_B = \arg\max_{c \notin \{c_A, R\}} \mathbb{P}(f_R(x + \delta) = c)$. Since we have

$$p_A + \sum_{c=1,..,K,\ c \notin \{c_A, R\}} p_c + p_R = 1 \tag{6}$$

it holds for $c_B = \arg\max_{c \notin \{c_A, R\}} \mathbb{P}(f_R(x + \delta) = c)$ that

$$\mathbb{P}(f_R(x + \delta) = c_B) = 1 - p_R - p_A - \sum_{c \notin \{c_A, c_B, R\}} p_c \leq 1 - \underline{p_{A+R}}$$

and so the right-hand-side of the inequality serves as an upper bound on $p_B$. Substituting this into Eq. 5 one can get

$$R \geq \frac{\sigma}{2}(\Phi^{-1}(\underline{p_A}) - \Phi^{-1}(\overline{p_B})) \geq \frac{\sigma}{2}(\Phi^{-1}(\underline{p_A}) + \Phi^{-1}(\underline{p_{A+R}}))$$

where we have used $\Phi^{-1}(1 - z) = -\Phi^{-1}(z)$. $\qquad \square$

## 6.2 Training the rejectors

In this work, we have assumed that a pretrained classifier along with an affordably pretrained denoiser are given to enable robustness and certification of the given classifier. The goal is to utilize rejectors $\{h_k\}_k$ with inexpensive/affordable training to increase the robustness certification of the joint system.

As the proposed procedure suggests, an ideal rejector $h_k()$ is the one that can successfully discriminate between the correctly classified and mis-classified denoised inputs that are assigned to class $k$ by the base classifier $f(.)$. Thus, we propose to train these networks according to this objective. Concretely, with $\ell_{BCE}$ denoting binary cross-entropy loss, define the classification loss for rejector $h_k$ as

$$L_{\Omega_k}(x_i, \epsilon) = \ell_{\text{BCE}}(H_k(D(x_i + \epsilon)), b_i) \tag{7}$$

where $H_k$ is the sigmoid outputs of rejector $k$, and the target label $b_i$ for image $x_i$ is defined as

$$b_i := \mathbb{1}_{\{f(D(x_i+\epsilon)) \neq f(x_i)\}} \tag{8}$$

where $\mathbb{1}_{\{.\}}$ is the indicator function. That is $b_i = 0$ if the classifier $f$ has classified denoised input $D(x_i + \epsilon)$ to the same class as that of the noise-free image $f(x_i)$, and $b_i = 1$ otherwise, thus rejecting

the noisy images whose classification outcome has changed. [2] Capturing the set of parameters of all the $K$ rejectors by $\Omega := \{\Omega_1, \cdots, \Omega_K\}$, total loss aggregated over the entire set of data with all possible $K$ classes yields

$$L_{\text{rejection}}(\Omega) := \mathbb{E}_{x_i, \epsilon} \sum_{k=1}^{K} 1_{f(x_i)=k} \; L_{\Omega_k}(x_i, \epsilon) \; . \tag{9}$$

In order to make the training more affordable, we propose to tie the $K$ rejectors through a shared backbone $h_{\text{BB}}$, parameterized by $\Omega_{\text{BB}}$ and define each $h_k$ by adding a fully-connected layer parameterized by $\omega_k$ to the features extracted via the backbone network; see Fig. 2. This renders the minimization of $L_{\text{rejection}}$ w.r.t. $\Omega = \{\Omega_{BB}, \omega_1, \cdots, \omega_K\}$ upon approximation by empirical mean over the dataset and $T \geq 1$ realizations of Gaussian noise to simplify to

$$\min_{\{\Omega_{\text{BB}}, \omega_1, \cdots, \omega_K\}} \mathbb{E}_{x_i, \epsilon} \sum_{k=1}^{K} 1_{\{f(x_i)=k\}} \; L_{\Omega_{\text{BB}}, \omega_k}(x_i, \epsilon) \equiv$$

$$\min_{\{\Omega_{\text{BB}}, \omega_1, \cdots, \omega_K\}} \frac{1}{NT} \sum_{i=1}^{N} \sum_{t=1}^{T} \sum_{k=1}^{K} 1_{\{f(x_i)=k\}} \; L_{\Omega_{\text{BB}}, \omega_k}(x_i, \epsilon_t).$$

### 6.3  Experiment details

**Randomized smoothing with training classifiers against Gaussian perturbations.** The work of [Cohen et al., 2019] obtains the classifier by training its weights with augmented input images with Gaussian noise. This method serves as the upper-bound baseline in robustifying pretrained networks, as the pretrained classifier's weights are given and cannot be altered while randomized smoothing optimizes those.

**Denoised smoothing**. [Salman et al., 2020] uses a trainable denoiser and trains it with the (i) MSE loss that is fast to train (0.78 to 3 hours of training), (2) MSE loss then fine-tuning by the 'classification-stability' loss (total of 1 to 3.8 hours of training), and (3) the 'classification-stability' loss (9.8 to 20.8 hours of training). While the last objective gives the highest performance, it takes almost an order of magnitude longer to train compared to other objectives, with up to training 600 epochs as proposed by the authors. Two choices of recent denoiser architecture, namely DnCNN [Zhang et al., 2017] and MemNet [Tai et al., 2017], have been tested, the choice of which influences the training time. We utilize the weights shared by the authors under MIT License and take the best performing selection of training hyperparameters per denoiser architecture and noise parameter $\sigma$.

**Our method**. In this setup, we assume that the classifier and denoiser architectures and weights are given and cannot be changed. In order to verify the effectiveness of the proposed method with sample rejection, we augment the system consisting of the classifier and denoiser with a rejector network $h_\Omega(.)$ with a ResNet-34-like backbone architecture, followed by 3 layers of fully connected (FC) layers with a 10-dimensional output, to realize the $K = 10$ rejector heads[3], given the 10 classes in the CIFAR10 dataset. Additionally, we have adjusted the architecture of the backbone slightly so that mid-layer features of the classification network $f(.)$ are concatenated with the mid-layer features of the rejector right after each of the 4 "blocks" in the Resnet architectures, and are fed forward into the rejector network. We have found that explicit use of classifier features yields empirical advantage, while it is important to note that the gradient does not flow backwards into the classifier architecture as the weights of network $f(.)$ are fixed.

We train the $h_\Omega(.)$ network with a total of 20 epochs where the first 5 epochs utilize Adam optimizer with $1e-4$ learning rate, followed by an SGD optimizer that starts with learning rate of 1e-3 and drops by a factor of 10 every 5 epochs. We train a rejector for every denoiser given by denoised-smoothing [Salman et al., 2020] as discussed above.

---

[2]It is interesting to note that there are similarities between this loss definition and that of the classification stability $L_{stability}$ in 2.2. One could also define target $b_i$ according to the ground-truth class $y_i$, however, this would impose a harder criteria, which is more difficult to aim for, as also observed in the training the denoiser in [Salman et al., 2020].

[3]One should consider the first two layers of the 3-layer FC network as part of the backbone network, and view the last FC layer as the independently trained $K$ heads

**Certification accuracy and training time**

For a given (classifier $f$, denoiser $D$, rejector $h_\Omega$), the certified radii of CIFAR10 test set are calculated using the Theorem 3.1 for $\alpha = 0.01$ and $N = 10,000$, and the certification curves are plotted by calculating the percentage of the data points whose radii are larger than the given $\ell_2$-radius.

The experiments are carried out for $\sigma \in \{0.25, 0.50, 1.00\}$, and the results are plotted in Figures 4 and 5 for DnCNN and MemNet denoiser architectures, respectively. For higher resolution copies of these plots together with additional choice of $\sigma = 0.12$ see Appendix 6.4. We chose $T = 20$, leading to a per-epoch training-time of 350 seconds for the DnCNN-based denoisers and 540 seconds for the MemNet-based denoisers, respectively; See Table. (1) for detailed training-time comparisons, and 9 for ablation study on $T$. Fig. 4 and 5 demonestrate that for both denoiser architectures, the proposed method with rejection has better certification accuracy (except for $\sigma = 1.0$ with MemNet and MSE objective). With $\sigma$ being 0.25 or 0.5, the gain of certified accuracy with rejection is more significant especially at radius 0. This shows that by adding the rejector, the clean accuracy of the smoothed classifier $g_R(.)$ in Eq. (2) is better than the base classifier $f(.)$. Such increase of clean accuracy also helps increase certification radii since larger $\underline{p_{A+R}}$ will lead to larger radii as in Theorem 3.1.

Since the proposed randomized smoothing with sample rejection utilizes pretrained denoisers, making direct comparison difficult, we emphasize that the point to demonstrate through empirical results is the additional gain that can be obtained by utilizing a reject class on a denoised classifier. Our method can specifically improve cheaply trained denoisers and improve their performance without imposing a large computational cost; see Table (1) for training-times.

Although certification accuracy of our method is lower than denoised smoothing baseline with stability objectives, our training time is less than half of training a denoiser with stability loss – training a denoiser with MSE or MSE and stability objective in addition to train the rejector is in total 2.78 to 6.8 hours while training a denoiser with stability objective is 9.8 to 20.8 hours.

Furthermore, Fig. 6 plots the 2D-histograms corresponding to the overall denoiser-classifier-rejector system in terms of number-of-rejected vs. top-class not-rejected samples for the publicly available DnCNN denoiser architecture trained with the MSE-loss and fine-tuned with the stability loss, for various values of $\sigma$. The plots shows that indeed, by utilizing the rejector, as expected and discussed in Figure 1 and Remark in Section 3.1,, the total number of the samples in the top-class (not-rejected) and the reject-class lie close to the diagonal line $x_1 + x_2 = N$ where $N = 10,000$, especially for larger $\sigma$, giving high values for the (lower-bounds of) $\underline{p_{A+R}}$, leading to an improved certified radius.

### 6.4 Certification radius plots

Figures 7 and 8 plot the results on CIFAR10 as discussed in Section 4 with higher resolution together with the additional choice of $\sigma = 0.12$ (which was omitted from the main body due to space limitation).

### 6.5 Experiments- Ablation on varying $T$

Ablation study on number of noise realizations $T$ in empirical approximation of the training loss for training the rejector is presented here in Fig. 9. For these experiments, the best DnCNN-based denoiser from [Salman et al., 2020] trained with the MSE-loss and fine-tuned by the stability loss was selected. training parameters of the rejectors is selected as explained in Section 5 while $T \in 4, 6, 10, 20$.

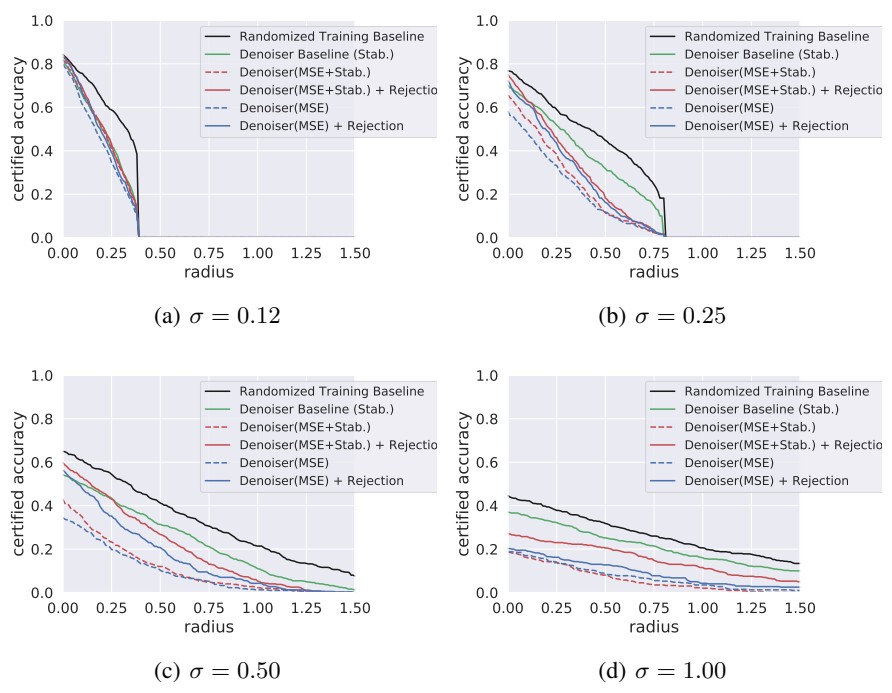

(a) $\sigma = 0.12$                      (b) $\sigma = 0.25$

(c) $\sigma = 0.50$                      (d) $\sigma = 1.00$

Figure 7: CIFAR10 certification results with DnCNN-based denoising

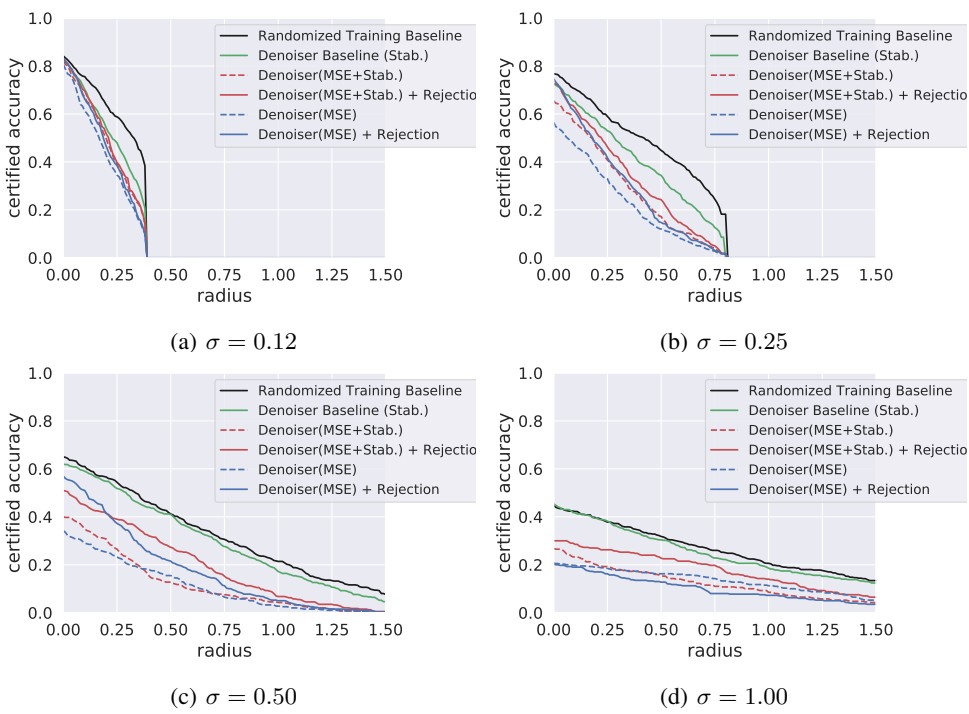

(a) $\sigma = 0.12$                      (b) $\sigma = 0.25$

(c) $\sigma = 0.50$                      (d) $\sigma = 1.00$

Figure 8: CIFAR10 certification results with MemNet-based denoising

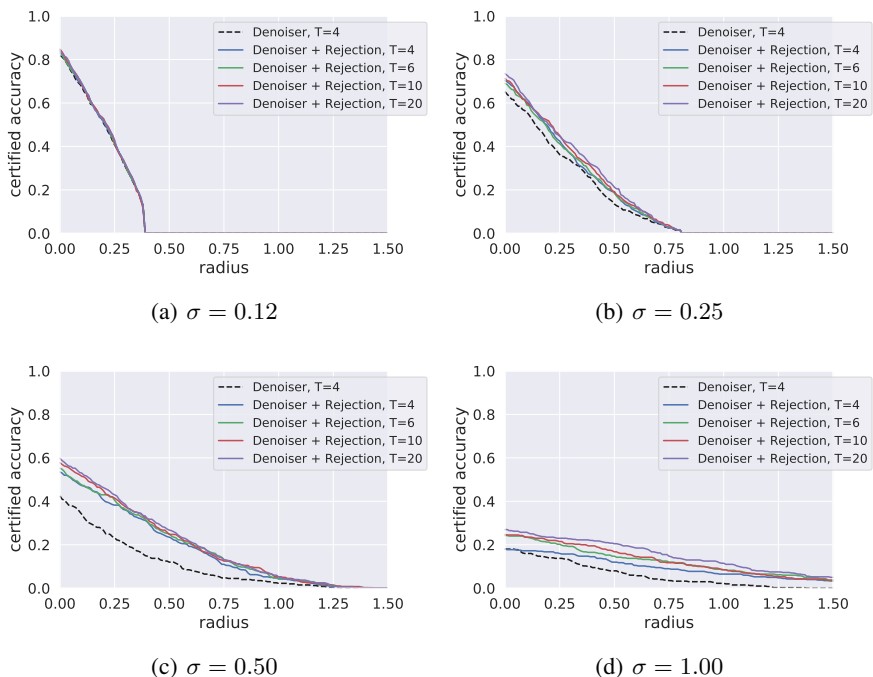

(a) $\sigma = 0.12$

(b) $\sigma = 0.25$

(c) $\sigma = 0.50$

(d) $\sigma = 1.00$

Figure 9: Ablation study on number of noise realizations $T$ in empirical approximation of the training loss.

