# OpenReview forum: "Denoised Smoothing with Sample Rejection for Robustifying Pretrained Classifiers"
_NeurIPS.cc/2022/Workshop/TSRML — TSRML2022_

### Official Review · Reviewer_KoFZ · 2022-10-20

**Overall Recommendation:** See above.
**Overall Rating:** 5

**Summary:**

This paper focuses on black-box certification against $\ell_2$-norm adversarial perturbation via randomized smoothing. In particular, the paper proposed adding a new rejection class to a base classifier.

**Strengths:**

+ The idea is novel.
+ Evaluations on benchmark datasets.
+ The proposed method is more efficient than baselines.

**Weaknesses:**

- The certified accuracy of the proposed method is lower than the baseline.

**Review Confidence:**

3: The reviewer is fairly confident that the evaluation is correct

---

### Official Review · Reviewer_nY7w · 2022-10-20
**A good direction to imporve certified acc**

**Overall Rating:** 7

**Summary:**

This paper proposes to utilize a sample rejection for noisy examples, which is able to further improve the performance of certified defense (denoised smoothing).

**Strengths:**

This paper is the first to propose the use of sample rejection on certified defense. In addition, good writing and figures help readers easier to read this paper.

**Weaknesses:**

Need more experiments and experimental analysis on different datasets (e.g. imageNet dataset) to demonstrate the efficacy of the proposed method.

**Overall Recommendation:**

An interesting direction is to improve the performance of the certified defense, however, it needs more experiments to demonstrate its efficacy.

**Review Confidence:**

4: The reviewer is confident but not absolutely certain that the evaluation is correct

---

### Official Review · Reviewer_xSbc · 2022-10-21
**Interesting work for introduing sample rejection to denoised smoothing**

**Overall Rating:** 7

**Summary:**

Generally, the prediction accuracy of the pretrained classifier on the denoised images will drop slightly compared to the clean accuracy. To tackle this problem, this work proposes to add one more reject class on the base classifier, which is realized through several newly trained per-class rejectors. Then, by leveraging this reject class, the estimation of the upper bound for the probability of the runner-up class can be tightened, and the certified accuracy under denoised smoothing will thus be further improved.

**Strengths:**

This work utilizes a rejection class to further improve the certified robustness under denoised smoothing, which is quite interesting and novel. The paper itself is well-written, the claims are well supported with the theoretical analysis, and the performance improvement is also significant under the medium or large magnitude of the smoothing noise.





**Weaknesses:**

+ The current result is not so good compared to the baseline that uses the denoiser trained with stability objective. I suggest the author add the experiment result for the Stab + Reject for completeness and better comparison. The time for such finetuning (Stab + Reject) should not be expensive compared to the original training for the Stab objective.

+ The proposed method may not be quite scalable. With the Reject Obj, the training time for each epoch is about six to ten times larger than the baseline, and the number of rejector heads K here is only 10. Then, I doubt it will be pretty hard to scale this method to a larger dataset like ImageNet with K = 1000.

+ The time cost for the certification with the Reject compared to the baseline should also be reported and discussed.

+ A minor thing is that the confidence for the certification should be set as the same for both baseline and the proposed method. For the baseline, the confidence is $(1-\alpha)$, while for the proposed method, it is actually $(1-\alpha)^2$ instead of $(1-\alpha)$.



**Overall Recommendation:**

Although the result is a bit incomplete, and there is still a gap between the performance of the proposed method and the performance of the baseline Stab. However, the overall idea is novel, and the experiment results based on other baselines are enough to show the potential effectiveness of the proposed method; I would recommend an acceptance of this work.


**Review Confidence:**

3: The reviewer is fairly confident that the evaluation is correct

---

### Decision · Program_Chairs · 2022-10-23

**Decision:**

Accept

**Comment:**

This submission is accepted following the majority of recommendations. We encourage the authors to incorporate the detailed reviewer feedback, e.g., by adding more discussion on the time cost, scalability, and confidence computation of the proposed method, and by comparing with "Stab + Reject" and evaluating on ImageNet for completeness, in the camera-ready version.